# A Novel Combination of Sotorasib and Metformin Enhances Cytotoxicity and Apoptosis in KRAS-Mutated Non-Small Cell Lung Cancer Cell Lines through MAPK and P70S6K Inhibition

**DOI:** 10.3390/ijms24054331

**Published:** 2023-02-22

**Authors:** Pedro Barrios-Bernal, José Lucio-Lozada, Maritza Ramos-Ramírez, Norma Hernández-Pedro, Oscar Arrieta

**Affiliations:** Thoracic Oncology Functional Unit (UFOT), Laboratorio de Medicina Personalizada, Instituto Nacional de Cancerología, S.S.A., San Fernando 22 Sección XVI, Tlalpan, Mexico City 14080, Mexico

**Keywords:** sotorasib, metformin, KRAS, AKT, P70S6K, MAPK, lung cancer

## Abstract

Novel inhibitors of KRAS with G12C mutation (sotorasib) have demonstrated short-lasting responses due to resistance mediated by the AKT-mTOR-P70S6K pathway. In this context, metformin is a promising candidate to break this resistance by inhibiting mTOR and P70S6K. Therefore, this project aimed to explore the effects of the combination of sotorasib and metformin on cytotoxicity, apoptosis, and the activity of the MAPK and mTOR pathways. We created dose–effect curves to determine the IC50 concentration of sotorasib, and IC10 of metformin in three lung cancer cell lines; A549 (KRAS G12S), H522 (wild-type KRAS), and H23 (KRAS G12C). Cellular cytotoxicity was evaluated by an MTT assay, apoptosis induction through flow cytometry, and MAPK and mTOR pathways were assessed by Western blot. Our results showed a sensitizing effect of metformin on sotorasib effect in cells with KRAS mutations and a slight sensitizing effect in cells without K-RAS mutations. Furthermore, we observed a synergic effect on cytotoxicity and apoptosis induction, as well as a notable inhibition of the MAPK and AKT-mTOR pathways after treatment with the combination, predominantly in KRAS-mutated cells (H23 and A549). The combination of metformin with sotorasib synergistically enhanced cytotoxicity and apoptosis induction in lung cancer cells, regardless of KRAS mutational status.

## 1. Introduction

KRAS mutations occur in up to 35% of patients with non-small cell lung cancer (NSCLC) [1] and represent 50% of oncogenic mutations in adenocarcinoma histology. Clinically, these genetic alterations are usually related to age over 65 years, smoking history, mutual exclusivity from alterations in the Epidermal Growth Factor Receptor (EGFR), and EML4-ALK translocations [2,3]. Furthermore, alterations in the KRAS oncogene have been considered predictors of poor response in chemotherapy-treated NSCLC patients harboring advanced, or metastatic, disease stages [4]. The biological importance of KRAS mutations is focused on impairing GTP hydrolization, keeping it aberrantly activated [5], which then results in constitutive activation of cell signaling pathways, such as mitogen-activated protein kinase (MAPK) and AKT-mTOR-P70S6K [6]. In NSCLC, these alterations occur mainly at codon 12 (80%), mostly as a substitution of glycine by cysteine (G12C, 42%), but there are also reported interchanges of glycine for valine (G12V, 21%), glycine for aspartate (G12D, 17%), and glycine for alanine (G12A, 7%) [7]. Particularly, G12C mutation is relevant, as it binds to specific KRAS inhibitors, such as sotorasib and adragasib [8], then inhibiting the phosphorylation of p-ERK in cells with this mutation [6], correlating with important reductions in tumoral size [5,6], and even showing promising results in clinical trials [9,10] of lung, colorectal, pancreatic, and endometrial cancers [11]. However, although the antineoplastic effects of sotorasib have been clearly described, their short-lasting clinical responses have become its most important drawbacks [10,11,12]. Consequently, preclinical evidence has suggested that sotorasib efficacy in KRAS-mutated tumors may be affected by diverse off-target resistance mechanisms [13], among which the most relevant is MAPK pathway reactivation by AKT-mTOR-P70S6K signaling [14]. Thus, metformin represents a pharmacological alternative that may overcome this resistance mechanism, since this biguanide inhibits complex 1 of the mitochondrial respiratory chain, subsequently activating AMP-activated protein kinase (AMPK). This triggers the activity of intracellular intermediaries to inhibit mTORC1, finally decreasing proteinic synthesis in cancer cells through p70S6K inhibition [15]. Accordingly, diverse studies have evidenced the cytotoxic role of metformin as monotherapy, its capacity to promote apoptosis and inhibit the mTOR pathway, as well as the correlation of these findings with reduced tumoral sizes in murine models [16,17,18].

Additionally, the combination of metformin with tyrosine kinase inhibitors, like afatinib, synergistically increased cell cytotoxicity, induction of apoptosis, and inhibition of PI3K-AKT-mTOR pathway in A549 (KRAS G12S) cell lines, even if these cells lack of EGFR mutations. This further suggests a sensitizing effect on afatinib mediated by metformin [19], which was further supported by in vivo studies reporting that combining metformin with diverse other targeted therapies decreased tumoral size and inhibited mTOR signaling in mouse neoplasms derived from A549 cell line (KRAS G12S) [20].

As well, in vitro evidence has demonstrated that metformin also regulates MAPK pathway; for instance, Ko et al. [17] showed that increasing metformin concentrations exhibited a dose-dependent inhibition of p-MEK1/2 and p-ERK1/2 in A549 and H1975 cells. Comparably, Do et al. [16] identified that metformin inhibited p-Raf and p-ERK1/2 in a dose-dependent manner. Thus, the effects of this biguanide extend beyond mTOR pathway.

Furthermore, emerging clinical evidence supports the concomitant use of metformin with antineoplastic drugs; for example, Arrieta et al. [21] reported that the combination of metformin with tyrosine kinase inhibitors (TKIs) increased the overall and progression-free survival periods in patients with EGFR-mutated NSCLC. Similarly, a phase II clinical trial showed a significant increase in the progression-free survival (PFS) of NSCLC patients after combined treatment with metformin and paclitaxel, carboplatin, or bevacizumab [22]. These findings suggest that metformin may enhance the clinical effectivity of other antineoplastic agents [20,23,24,25].

Finally, the molecular consequences derived from combining metformin and sotorasib remain unexplored; therefore, this study aimed to analyze their effects on cell viability, apoptosis and the activity of MAPK and AKT-mTOR pathways in lung cancer cell lines harboring different KRAS mutational statuses.

## 2. Results

### 2.1. Metformin Increases Sotorasib-Driven Cytotoxicity in KRAS-Mutated Lung Cancer Cell Lines

First, we found a greater decrease in cellular viability using the combination of metformin and sotorasib, compared to their corresponding monotherapies. Specifically, we found significant differences between the combination and sotorasib alone in KRAS-mutated cell lines H23 (56.2% vs. 44.6%; *p* = 0.0457; Figure 1A) and A549 (31.6% vs. 53.9%; *p* = 0.0223; Figure 1B). Differently, the wild-type KRAS cell line (H522) did not display statistical significance in this comparison (57.4% vs. 47.6%; Figure 1C). Moreover, the pharmacodynamic analysis reported synergy between sotorasib and metformin in all tested cell lines, including H23 (CI = 0.62450; Figure 1A), A549 (CI = 0.73647; Figure 1B), and H522 (CI = 0.91655; Figure 1C).

### 2.2. Increased Apoptosis Induction by the Addition of Metformin to Sotorasib, Regardless of KRAS Status

After, we measured membrane markers of apoptosis (annexin-V) or necrosis (7-AAD). Consequently, as shown in Figure 2, all cell lines exhibited increases in apoptosis induction driven by the combination, compared to controls, including H23 (22.3% vs. 70.27% *p* ≤ 0.0001), A549 (8.02% vs. 80.99% *p* ≤ 0.0001), and H522 cells (1.6% vs. 49.47% *p* ≤ 0.0001). Particularly, sotorasib showed significant differences compared to controls in H23 (24% vs. 66.2% *p* ≤ 0.0001) and A549 cells (5.6% vs. 71.7% *p* = 0.0127). Differently, H522 was the only cell line showing notable differences between sotorasib and the combination (64.5% vs. 80.9% *p* = 0.0217).

### 2.3. Combined Therapy Significantly Decreases MAPK Pathway Activity

After confirming that metformin and sotorasib concomitantly induced cell death, we assessed their biological impact on diverse intermediaries of MAPK pathway, such as KRAS, CRAF, BRAF, and ERK1/2. As expected, KRAS expression was importantly reduced in H23 cells after treatment with sotorasib alone (*p* = 0.0103) or in concomitance with metformin (*p* = 0.0013). In A549 cells, p-CRAF was importantly inhibited by all treatments, while BRAF expression was only reduced in the combined group (*p* ≤ 0.01). Additionally, p-CRAF was inhibited by the combined treatment in H522 cells (*p* ≤ 0.05). Furthermore, p-ERK1/2 (p-MAPK) expression was decreased in H23 by all treatments, in A549 cells by metformin (*p* ≤ 0.01) and the combination (*p* ≤ 0.01), and in H522 by the combination (*p* ≤ 0.01) and metformin alone (*p* ≤ 0.01) (Figure 3).

### 2.4. Combined Treatment of Metformin and Sotorasib Inhibits AKT and P70S6K Activation

Next, we explored the inhibitory efficacy of the combination over the AKT-mTOR-P70S6K pathway, since this is the main resistance mechanism to KRAS inhibitors (Figure 4). Specifically, AKT expression was reduced after sotorasib alone (*p* ≤ 0.05) or the combination (*p* ≤ 0.01) in H522 cells. Moreover, p-AKT was significantly inhibited by the combination in H23 (*p* = 0.0163) and H522 cells (*p* ≤ 0.05), but only as a non-significant trend in A549 cells. Furthermore, p-P70S6K was significantly inhibited by the combination in H23 (*p* = 0.0071) and H522 cells (*p* ≤ 0.01), but only as a non-significant trend in A549 cells.

## 3. Discussion

Treatment with sotorasib has modified response and survival of patients with KRAS G12C mutations. However, despite showing promising responses, intrinsic or acquired resistance mechanisms have prevented the development of better clinical results. In this sense, the most important mechanism of resistance to sotorasib is the activation of the AKT-mTOR-P70S6K pathway. As metformin has previously been demonstrated to inhibit this signaling pathway, we explored whether combining this biguanide with sotorasib resulted in an improvement of sotorasib effectiveness in lung cancer cells. Consequently, our results exhibited that the combination exerted synergistic effects over cytotoxicity and apoptosis in cells with G12C and G12S KRAS mutations. The most similar example to this phenomenon in the literature is a study of our research group, showing that combining metformin and afatinib (EGFR tyrosine kinase inhibitor) induces a synergistic effect on A549 cells, even if this cell line lacks EGFR mutations. This effect was mainly attributable to metformin-driven AMPK activation, which then inhibited mTOR-P70S6K signaling [19]. Analogously, metformin also potentiates apoptosis in combination with selumetinib (MEK inhibitor) [26], implying that the inhibition of the MAPK pathway is important for metformin-driven apoptosis as part of a wide mosaic of other reported mechanisms, such as lowering of Bcl-2 protein levels, increasing Bax expression [27], and promoting G0/G1 cell cycle arrest [18]. Otherwise, sotorasib has also been combined with other drugs to overcome its resistance, like buparlisib (PI3K inhibitor) [28] or DT2216 (BCL-XL) [29], thereby supporting the assertion that the PI3K-AKT-mTOR pathway plays an important role in the apoptosis of KRAS-mutated cells [30].

After assessing the cytotoxic effects of our concomitant therapy, we explored its impact on MAPK and AKT-mTOR-P70S6K signaling pathways, showing an important inhibition of them in all cells tested, regardless of KRAS mutational status. This is relevant, since MAPK pathway inhibition is a well-known consequence of sotorasib monotherapy in models with G12C mutation [31], and it is equally expected that its high specificity to this alteration prevents sotorasib from inhibiting p-ERK in cells without G12C mutation [6]. Therefore, our results show, even at the proteomic level, an important sensitization of metformin to sotorasib effects in non-common KRAS mutations. Furthermore, mTOR inhibition has special importance in reaching effective cytotoxicity in cells with an over-activated MAPK pathway, as important cytotoxic effects in cell lines with KRAS or MEK mutations are reported from the use of mTOR inhibitors, whether alone [32,33] or in combination with MAPK inhibitors [34]. Finally, we found that metformin synergizes with sotorasib due to an important inhibition of AKT and P70S6K in all cells. These findings match with those results previously reported by our research group for combining metformin and afatinib in lung cancer cells, in which we described that this biguanide potentiates apoptosis induction by inhibiting the EGFR-AKT-P70S6K pathway [19]. Furthermore, previous studies have also reported that inhibiting PI3K-AKT-mTOR pathway positively correlated with apoptosis induction [32,33]. These findings are further consistent with preclinical evidence testing the concomitant use of metformin and figitumumab, showing inhibition of PI3K-AKT and MAPK signaling pathways [20], thus placing these drugs as potential enhancers of KRAS inhibitors, such as sotorasib. Differently, metformin has demonstrated variable outcomes over MAPK, as some studies stand that this biguanide increases B-RAF and C-RAF activity [35,36], while others report the exact contrary effect [16,17]. This phenomenon can be explained by differences in the concentrations used during in vitro tests; for instance, metformin IC50 concentrations > 20 mmol in A549 cells are reported to cause an active inhibition of the AKT-mTOR pathway, which decreases the inhibitory activity of Rheb over the dimerization of C-RAF and B-RAF [35], indirectly promoting MAPK activity. Meanwhile, lower concentrations of this biguanide (1–10 mmol) are not reported to inactivate Rheb, then allowing MAPK pathway inhibition, as evidenced in this study for A549 (CRAF and P-MAPK), and H522 cells (P-CRAF, P-MAPK, CRAF, and MAPK) after treatment with metformin, either as a monotherapy or in combination with sotorasib. Altogether, our results suggest that combining metformin and sotorasib finds its main mechanism of action in the concomitant inhibition of the AKT-mTOR-P70S6K pathway by metformin, and MAPK by sotorasib, thus simultaneously decreasing protein synthesis and cell growth. This mechanism of action is further illustrated in Figure 5.

Moreover, as part of the wide mosaic of intracellular effects of metformin, plenty of evidence demonstrates that this biguanide modifies diverse metabolic pathways to avoid the development of Warburg effect in cells with KRAS mutations. Although we did not evaluate metabolism in this study, we previously reported that combining metformin and afatinib showed strong inhibition of GLUTs, and a marked increase in AMPK activity, regardless LKB1 involvement [19]. This may be explained by AMPK-driven inhibition of energy generation [37]. Importantly, our study shows that cells lacking LKB1, such as A549, decreased MAPK and p-MAPK expressions, which may also promote metabolic consequences, such as decreased lactate levels and AMPK-mediated glycolysis. Therefore, the metabolic importance of metformin may be of special interest in cells with KRAS mutations, as this driver alteration is metabolically involved in cancer progression [38].

### Strengths and Limitations

The main strength of this study is exploring the combined effect of metformin and sotorasib in cells having or not KRAS G12C mutation of susceptibility to sotorasib, demonstrating a synergistic relationship between metformin and sotorasib for the first time. Nevertheless, we are aware of the limitations of this investigation; first, it only was used one cell line belonging to each of the most representative groups of mutational profiles (KRAS G12C, G12S, and non-KRAS mutated). Second, although our results in A549 cells are in line with previous reports, evidence is lacking for H23 and H522 cells, not allowing complete generalization of our results to studies involving these cell lines.

## 4. Materials and Methods

### 4.1. Cell Lines and Reagents

Human lung adenocarcinoma cell lines H23 (KRAS G12C), A549 (KRAS G12S), and H522 (without KRAS mutations) were purchased from the American Type Culture Collection (ATCC, Manassas, VA, USA). H522 and H23 cells were cultured in RPMI-1640 medium (Gibco, Waltham, MA, USA. 31800-022), meanwhile A549 cells were cultured in F12 medium (Gibco, Waltham, MA, USA. 21700-075), and both media were supplemented in a 10% concentration with Fetal Bovine Serum (FBS) (Gibco, New York, NY, USA. 26140-079) and penicillin–streptomycin–amphotericin B in a 1% concentration (MP Biomedicals. Fountain Pkwy, OH, USA, 091674049). They were incubated in an atmosphere of 5% CO_2_ at 37 °C. As cells constituted an 80–90% confluent monolayer, they were subcultured using 400 µL of Trypsin-EDTA 1X solution (Sigma Aldrich. St. Louis, MO, USA. 549430C). 

Metformin (Sigma Aldrich. St. Louis, MO, USA. PHR1084) was diluted in the appropriate culture medium of each cell line at a concentration of 100 mmol. Similarly, sotorasib (Medkoo Biosciences. Morrisville, NC, USA. 207085) was diluted in dimethyl sulfoxide (DMSO) at 5 µmol, 10 µmol, 15 µmol, 20 µmol, and 25 µmol concentrations.

### 4.2. Cell Viability Assay

A quantity of 1 × 10^4^ cells per well were seeded in triplicate in 96-well plates. After 24 h of incubation, cells were treated for 72 h with metformin at different concentrations per well triplicate (5 mM, 10 mM, 15 mM, 20 mM, and 25 mM). In the same way, cells in three independent experiments were treated for 72 h with 5 µmol, 10 µmol, 15 µmol, 20 µmol and 25 µmol of sotorasib as monotherapy.

Subsequently, the MTT solution (3,4,5-dimethylthiazol-2-yl-2,5-didiphenyltetrazolium bromide) at a concentration of 5 mg/mL (Sigma Aldrich. St. Louis, MO, USA. Catalog number: M2128) was added to wells, which were incubated for 4 h. After this period, the culture medium was removed and replaced by 200 µL of isopropanol-DMSO (1:1) solution to dissolve the formazan crystals. Cell viability resulting from this experiment was quantified by measuring absorbance at 570 nm (BioTek, Saint Clare, CA, USA, ELX 808) to calculate optical density values.

The results of such measurements were averaged and normalized at 100% in relation to controls. According to cytotoxicity results, we determined IC50 and IC10 doses of sotorasib and metformin, respectively, for each cell line, which are shown in Table 1.

Then, each cell line was seeded in 96-well plates in an amount of 1 × 10^4^ cells per well, ordering them in five triplets of wells, representing the following treatment groups: control, DMSO, metformin, sotorasib and the combination of sotorasib and metformin. After 24 h of incubation, each cell line was treated with its respective IC10 and IC50 doses of metformin and sotorasib, respectively, either as monotherapy or as a combination. After that, the viability test was carried out using MTT solution and a spectrophotometer, as previously described.

### 4.3. Analysis of Drug Combination Index 

To determine the type of pharmacodynamic interaction between metformin and sotorasib, we calculated their combination index (CI) for each cell line using Compusyn 1.0 software (Biosoft, Cambridge, UK). Combination index values <1 were interpreted as synergistic, values from 1 to 1.10 as additive, and values > 1.10 as antagonistic.

### 4.4. Apoptosis Assay

To assess the level of apoptosis induction, cell lines were seeded in 24-well plates in a confluence of 4 × 10^4^ and incubated overnight. After 24 h, cells were incubated with IC10 and IC50 doses of metformin and sotorasib, respectively, either as monotherapies or as a combination for 72 h at 37 °C and 5% CO_2_. Then, the cells were detached using trypsin, washed three times with 1X PBS, and later they were marked with FITC Annexin V Apoptosis Detection Kit with 7-AAD (Biolegend, San Diego, CA, USA 640922). Finally, the cells were evaluated through flow cytometry in accordance with manufacturer’s instructions.

### 4.5. Western Blot Analysis

After 72 h of treatment, cell lines were washed three times with PBS solution and lysed with RIPA lysis buffer system (Santa Cruz Biotechnology. Dallas, TX, USA. SC-24948) according to the manufacturer’s instructions. Subsequently, extracted proteins were quantified using Bradford’s assay (Bio-Rad, Hercules, CA, USA, #5000205). Then, 40 µg of total protein was separated by electrophoresis for 110 min at 100 V on 10% SDS-PAGE gel, and then transferred onto 0.2 μmol nitrocellulose membranes by the Trans-Blot Turbo Transfer System set at 20 V and 2.5 amps. The efficacy of this process was checked by Ponceau Red stain. Subsequently, membranes were blocked with a 10% BSA-PBS tween solution for 30 min, underwent three washes of 10 min with PBS-Tween solution, and were incubated with their corresponding primary antibodies (dilution 1:1000) overnight at 4 °C. Secondary antibodies were directed against the following molecules: KRAS (Santa Cruz Biotechnology. Dallas, TX, USA. SC-30), B-RAF (Cell signaling. Danvers, MA, USA. 9433), C-RAF (Cell signaling. Danvers, MA, USA. 53745), p-CRAF (Cell signaling. Danvers, MA, USA. 9421), MAPK (Cell signaling. Danvers, MA, USA. 9102), p-MAPK (Cell signaling. Danvers, MA, USA. 4370), AKT (Santa Cruz Biotechnology. Dallas, TX, USA. SC-5298), p-AKT (Santa Cruz Biotechnology. Dallas, TX, USA. SC-514032), P70S6K (Cell signaling. Danvers, MA, USA. 9202), p-P70S6K (Cell signaling. Danvers, MA, USA. 9205), and GAPDH (Santa Cruz Biotechnology. Dallas, TX, USA. SC-47724).

After incubation, each primary antibody was removed from its corresponding membrane. Later, each membrane underwent three washes of 15 min with PBS-Tween solution. Once completed, membranes underwent incubation for 1 h with a 1:10,000 dilution of their corresponding secondary antibodies. After that, membranes took 5 of 10 min to reduce background derived from secondary antibodies. Finally, proteins of interest were visualized using an enhanced chemiluminescence kit (LI-Cor, Lincoln, NE, USA), and band intensities were quantified by densitometry using ImageJ software (1.49 version, National Institutes of Health, Bethesda, MA, USA).

## 5. Conclusions

Metformin exhibits a sensitizing role to sotorasib in non-KRAS-mutated cells. Furthermore, the combination of sotorasib and metformin exerts a synergistic enhancement of cytotoxicity and apoptosis induction, likely driven by the concomitant inhibition of MAPK and mTOR-P70S6K pathways in KRAS-mutated cells.

## Figures and Tables

**Figure 1 ijms-24-04331-f001:**
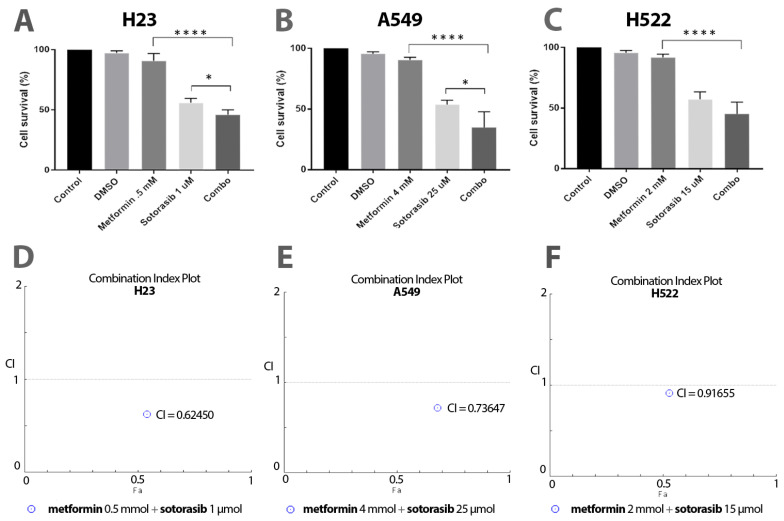
Cytotoxic effect of sotorasib and metformin as single agents or in combination. (**A**–**C**), MTT assays were performed to evaluate the cytotoxicity of these treatments. Each bar represents a mean of 3 independent experiments, and these drugs are tested in triplicate in each one. Two-way ANOVA was carried out to identify statistical differences among groups, indicated here as * *p* ≤ 0.05, **** *p* ≤ 0.0001. (**D**–**F**) combination index plots exhibited synergistic effects of combining metformin and sotorasib in all cells, described as points below 1 (threshold).

**Figure 2 ijms-24-04331-f002:**
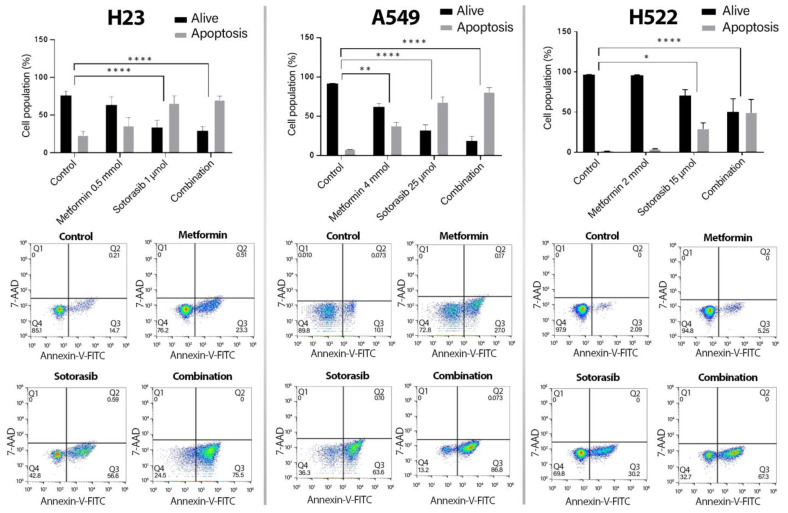
Induction of apoptosis by sotorasib and metformin as monotherapies or in combination. Cells were seeded and treated as previously described for 72 h. 5000 events were analyzed with the apoptosis kit and flow cytometry. Bars represent the means of 2 independent experiments in triplicate. Two-way ANOVA was carried out to perform the statistical analysis * *p* ≤ 0.05, ** *p* ≤ 0.01, **** *p* ≤ 0.0001. Q1, necrosis. Q2, late apoptosis. Q3, early apoptosis. Q4, living cells.

**Figure 3 ijms-24-04331-f003:**
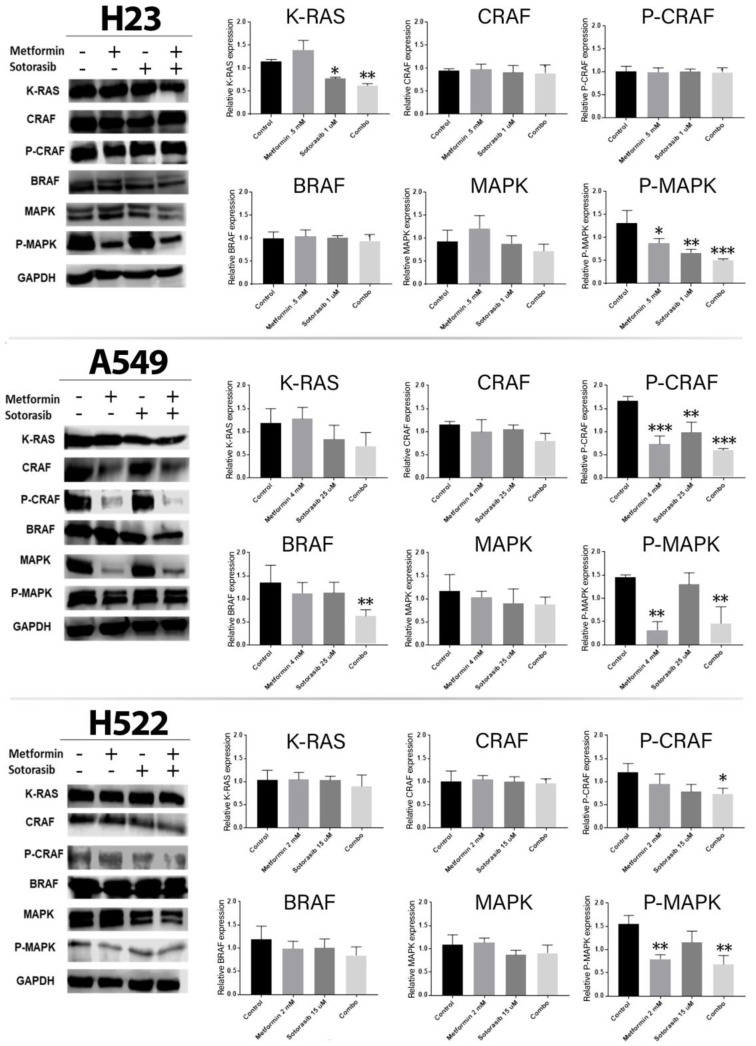
Effect of the combined treatment of metformin and sotorasib on the MAPK pathway. Cells were seeded and treated for 72 h with their respective therapy, after which protein was extracted and analyzed by Western blot. GAPDH was used as endogenous control, and Western blot images were analyzed by ImageJ software version 1.54 (NIH) and represented as bars in the graphics using GraphPad Prism software version 8.0.1 (Dotmatics, Boston, Massachusetts, USA). Each bar represents three independent experiments and results of the area are presented as mean and standard deviation. Data were normalized regarding endogenous control and statistically analyzed by one-way ANOVA. * *p* ≤ 0.05, ** *p* ≤ 0.01, *** *p* ≤ 0.001. MAPK and p-MAPK both refer to ERK1/2 and p-ERK1/2.

**Figure 4 ijms-24-04331-f004:**
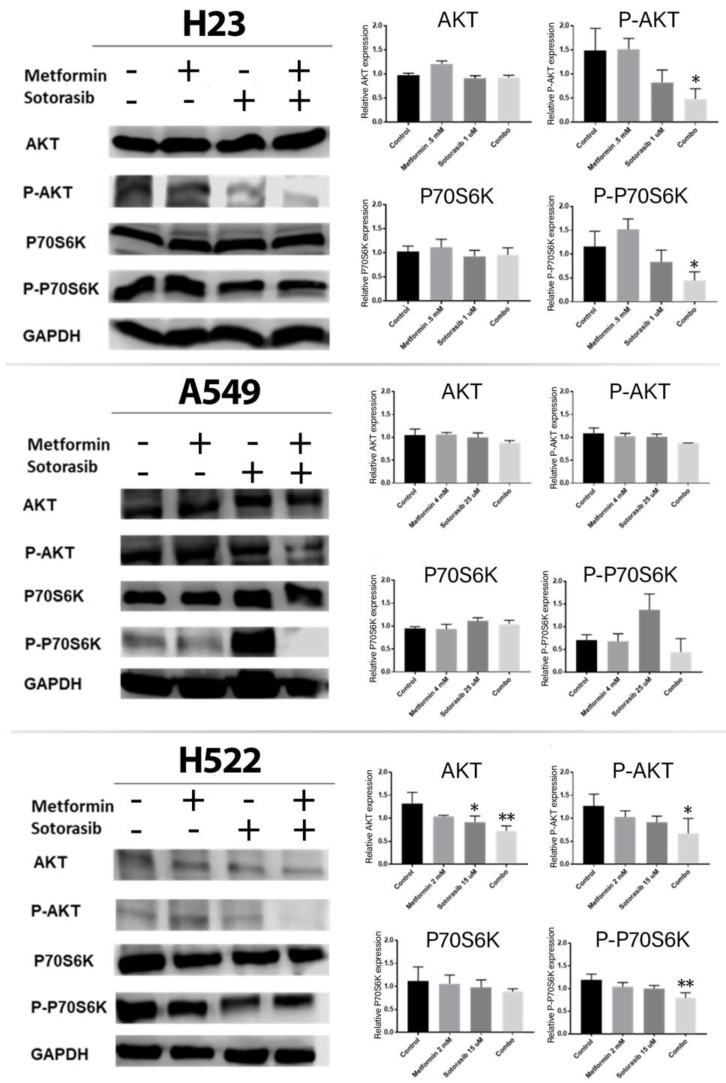
Effects of metformin and sotorasib on proteins involved in downstream signaling of growth factor receptors. Cells were seeded and treated for 72 h with their respective therapy. Then, protein was extracted and analyzed by Western blot using GAPDH as endogenous control. After this, resulting Western blot images were analyzed by ImageJ software version 1.54 (NIH) to quantify the intensity of each band. Subsequently, means of three independent experiments were represented as a single bar for each protein in the above graphics, created by GraphPad Prism software version 8.0.1 (Dotmatics, Boston, Massachusetts, USA). Data were normalized regarding GAPDH, and they were statistically analyzed using one-way ANOVA. Significant differences are represented as * *p* ≤ 0.05, ** *p* ≤ 0.01.

**Figure 5 ijms-24-04331-f005:**
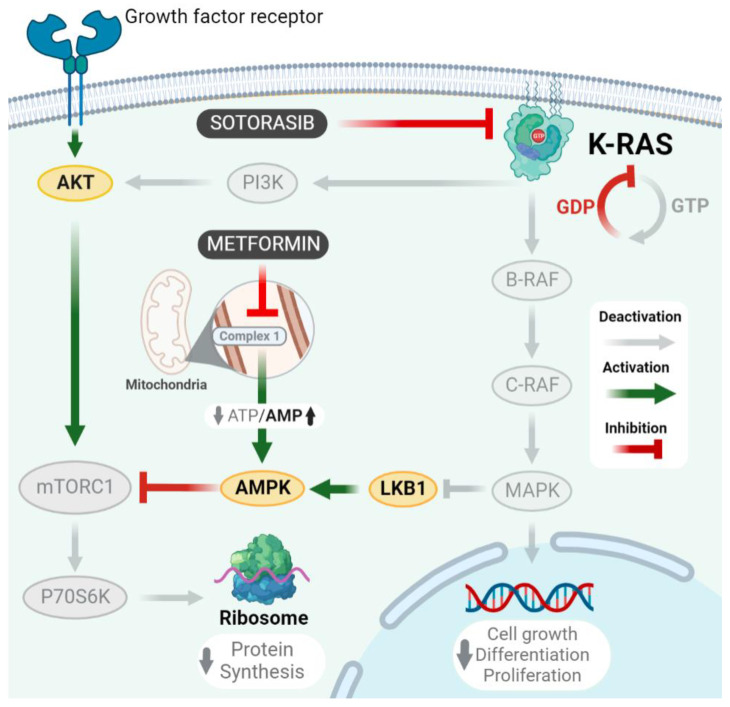
Possible action mechanism of metformin and sotorasib as combined treatment. Lung cancer cells enhance their proliferation and metastatic potential through mutations in growth factor receptors and KRAS. In detail, KRAS interacts with PI3K-AKT-mTORC1 pathway, and inhibits LKB1, then allowing mTOR-driven activation of P70S6K, which subsequently increases protein synthesis. As well, KRAS promotes the constitutive activation of MAPK pathway, enhancing cell growth and proliferation. In this context, our combined treatment simultaneously inhibits MAPK signaling by sotorasib, and P70S6K by metformin.

**Table 1 ijms-24-04331-t001:** Metformin and sotorasib work doses for each cell line.

Cell Line	Metformin (IC 10)	Sotorasib (IC 50)
A549	4 mmol	25 µmol
H522	2 mmol	15 µmol
H23	0.5 mmol	1 µmol

## Data Availability

No new data were created or analyzed in this study. Data sharing is not applicable to this article.

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
