# Peer review of "A Novel Combination of Sotorasib and Metformin Enhances Cytotoxicity and Apoptosis in KRAS-Mutated Non-Small Cell Lung Cancer Cell Lines through MAPK and P70S6K Inhibition"

_ijms, 2023, doi:10.3390/ijms24054331_

Round 1

Reviewer 1 Report

The manuscript by Barrios-Bernal et al is an interesting work on the activity of individual and in combination of sotorasib and metformin on small-cell lung cancer. The experiments and results are well justified. I have a minor suggestion. Some of the figures do look like as if they are stretched. Please fix them.

Author Response

Point. 1 The manuscript by Barrios-Bernal et al is an interesting work on the activity of individual and in combination of sotorasib and metformin on small-cell lung cancer. The experiments and results are well justified. I have a minor suggestion. Some of the figures do look like as if they are stretched. Please fix them.

Response 1. We took into account your comment, we improved the quality of the figures that are stretched. Additionally, we enhanced the quality of presented results.

Reviewer 2 Report

The manuscript by Bernal et al., aimed to use a combinational drug therapy to treat NSCLC by using metformin and Sotorasib. Though the aim is clear and straightforward, the authors have difficulty in their techniques especially with the western blots.

Major comments.

Fig.1. Why does author, select a different dose of Metformin and Sotorasib for different cell lines? Why there is no dose dependent effect of metformin or Sotorasib even with higher concentration on different cells?

Fig.2. How does the author not see any positive 7AAD cells on the FACS plot. Were there no dead cells, even after treatment? FACS plots are not convincing as different groups have different MFI. The author must consider acquiring a equal number of cells in the FACS especially if the absolute numbers are not shown.

Fig.3. What does MAPK mean? Is that p38, p44/42. How does the intensity of a phospho protein is higher than the total protein. Does the author calculate the relative intensity? The graph and image of the western blot do not match especially with MAPK and p-MAPK.

Fig.4. The author has used the same western blot images for both H23 and H522 groups, but the internal control is different. The graphs do not match with the blot images.

Most of the original blot images are highly contrasted.

Author Response

Point 1.  Fig.1. Why does author, select a different dose of Metformin and Sotorasib for different cell lines? Why there is no dose dependent effect of metformin or Sotorasib even with higher concentration on different cells?

Response 1. This is due to the fact that each cell line harbors a different response to each drug tested. Thereby, each cell line exhibits a different IC50 concentration for sotorasib and a distinct IC10 dose for metformin.

We performed dose-effect curves for both the monotherapy of treatments for each cell line, but these were not added to the manuscript.

Point 2.  Fig.2. How does the author not see any positive 7AAD cells on the FACS plot. Were there no dead cells, even after treatment? FACS plots are not convincing as different groups have different MFI. The author must consider acquiring a equal number of cells in the FACS especially if the absolute numbers are not shown.

Response 2. As 7-AAD and Annexin-V are markers of cell necrosis and apoptosis, respectively. As described in our results, we effectively observed cell death, but the main mechanism by which it occurred was apoptosis and not necrosis, thus this explains the lack of positivity to 7-AAD in our plots. 

We acquired an equal number of events (5,000 events) per group, as explained in the methodology section, in order to stablish objective conditions for the interpretation of our results. 

Point 3. Fig.3. What does MAPK mean? Is that p38, p44/42. How does the intensity of a phospho protein is higher than the total protein. Does the author calculate the relative intensity? The graph and image of the western blot do not match especially with MAPK and p-MAPK.

Response 3. MAPK pathway is an acronym of Mitogen-Activated Protein Kinase pathway, but some providers (Cell signaling. Danvers, MA, USA) name this antibody as “MAPK” (catalog number: 9102) referring to p44/p42 protein, as this is one of the last intermediates of the mentioned signaling pathway. 

We performed three independent experiments for each protein, then obtained three blot images per protein. After that, the relative intensities of the bands were determined by ImageJ software, and the means of these values were represented as bars in the graphics. Therefore, it is possible that blot images do not match with graphics, as doing this would be trying to compare the bands of one single experiment in the blot image, with the bars resulting from a mean of relative intensities in the graphics. 

Point 4. The author has used the same western blot images for both H23 and H522 groups, but the internal control is different. The graphs do not match with the blot images.

Most of the original blot images are highly contrasted.

Response 4. We appreciate this observation. We wrongly duplicated the blot images of H23 on H522, but this issue has already been solved by correctly placing the blot images of H522. 

We understand that a high-contrasted image may difficult its observation, but we augmented this parameter to enhance the definition of the bands.  

Additionally, we took into account your comments about the text and results, and we improved it.

Author Response

Point. 1 he current manuscript has limited mechanism in work and dual shutdown of MEK and PI3K/AKT
activation in responsive cells. In order to make it suitable for publication at IJMS, strengthening the MOA
is essential. Work is also needed to improve the scientific rigorousness of the manuscript.
In order to prove combo suppresses tumor growth through dual targeting of PI3K and MEK, the
authors need to provide significant amount of work benchmarking combo vs PI3Ki+MEKi, in vitro.
Overall a relatively superficial study lacking in vitro target engagement demonstration and limited
mechanistic insight.

Response 1. We required more information about the comments since we don't have targeted therapy for PI3K and MEK, our therapy is targeted to K-RAS, and metformin is not targeted therapy.

Reviewer 4 Report

The manuscript entitled A novel combination of sotorasib and metformin enhances cytotoxicity and apoptosis in KRAS-mutated non-small cell lung cancer cell lines through MAPK and P70S6K inhibition is presented for the peer review. mutations in KRAS harbor an important epidemiological and prognostic role due to the impairment of its ability to hydrolyze GTP, then resulting in constitutive activation of both pathways of mitogen-activated protein kinase (MAPK) and AKT-mTOR-P70S6K. Authors indeed pointed out that biological effect of the combined use of metformin with antineoplastic drugs wascorrelated with clinical evidence. This work aimed to explore the effects of the combination of these pharmacological agents on the viability and activation of MAPK and AKT-mTOR pathways in KRAS-mutated lung cancer cell lines. Paper is of great clinical and fundamental interest but has some issues.

First, line 92 is value 0.1132 shows  significance?

Second, have you applied pretreatment  PD184352 and U0126 as MAPK inhibitors?

Third, have you used qPCR for expression analysis?

Methods need clarification.

After resolving my issues, the paper will be acceptable.

Author Response

Point 1.  First, line 92 is value 0.1132 shows significance?

Response 1. Thank you for your comment, we solved this issue.

Point 2. Second, have you applied pretreatment  PD184352 and U0126 as MAPK inhibitors?

Response 2. No, we only used sotorasib, as the intention was to extrapolate its evidence as a combined treatment to in vivo and clinical contexts.

Point 3. have you used qPCR for expression analysis? 

Response 3. No, as we focussed on the effects of our combination on the proteinic expression of cell signaling intermediaries. 

Point 4. Methods need clarification.

Response 4. Suggested changes in this section were completed.

Additionally, we took into account your comments about the text and results, and we improved it.

Round 2

Reviewer 3 Report

The authors have addressed the questions

Reviewer 4 Report

Authors addressed all my comments and did great job. Thank you.